# An ultra energy-efficient hardware platform for neuromorphic computing enabled by 2D-TMD tunnel-FETs

Arnab Pal [1], Zichun Chai [1], Junkai Jiang[1], Wei Cao [1], Mike Davies[2], Vivek De[2] & Kaustav Banerjee [1] ✉

Brain-like energy-efficient computing has remained elusive for neuromorphic (NM) circuits and hardware platform implementations despite decades of research. In this work we reveal the opportunity to significantly improve the energy efficiency of digital neuromorphic hardware by introducing NM circuits employing two-dimensional (2D) transition metal dichalcogenide (TMD) layered channel material-based tunnel-field-effect transistors (TFETs). Our novel leaky-integrate-fire (LIF) based digital NM circuit along with its Hebbian learning circuitry operates at a wide range of supply voltages, frequencies, and activity factors, enabling two orders of magnitude higher energy-efficient computing that is difficult to achieve with conventional material and/or device platforms, specifically the silicon-based 7 nm low-standby-power FinFET technology. Our innovative 2D-TFET based NM circuit paves the way toward brain-like energy-efficient computing that can unleash major transformations in future AI and data analytics platforms.

Neuromorphic (NM) computing that mimicks certain neuro-biological architectures of the human brain, is an alternative to the conventional von Neumann computing architecture, and therefore, can be designed to be highly parallel and very low power consuming with the potential to perform certain complex operations faster and in a smaller footprint[1]. This promise of NM computing in eventually enabling brain-like energy-efficient computing, therefore, has led to explosive market growth recently (estimated to reach $35 billion by 2029[2]), particularly in application spheres of—mobile computing, IoT (estimated to produce an economic impact of up to $11 trillion by 2025; https://www.mckinsey.com/mgi/overview/in-the-news/by-2025-internet-of-things-applications-could-have-11-trillion-impact), and artificial intelligence (AI) applications, including language processing, image recognition, computer vision, robotics, etc., where NM can enable highly-efficient computing involving ultra energy-efficient devices. Additionally, NM computing, due to its integrated computational- and memory units, can also eradicate the memory wall[3] problem, which has been estimated to consume the majority of the energy and time in certain computation tasks and is further going to intensify in the age of big data[4].

Conventionally the most biologically plausible NM hardware implementations are based on spiking neural nets (SNN)[5–7], a third generation of artificial neural nets (ANN)[8,9], which represents biological neurons and their related synaptic weights in hardware, and therefore, by closely replicating the behavior of the human brain, are expected to be highly suitable for implementing machine learning (ML) based AI applications, which is of paramount importance in the current age of big data (Supplementary Note 1, Supplementary Fig. 1). Figure 1a shows such an interconnected biological connection of three neurons (A, B, and C) along with their synaptic connections, where spikes generated on the axons of neuron A(B) are transmitted across the synapse to the dendrite terminal of the next neuron B(C), raising their membrane potential and eventually causing them to fire once the potential exceeds the firing threshold. The subsequently generated output spike drives further firing events in other interconnected neurons, thereby representing the mechanism of neuronal communication.

However, despite decades of research into finding a suitable hardware platform for efficiently implementing SSN-based NM computing, the search has proved to be elusive. While NM computing can

[1]Department of Electrical and Computer Engineering, University of California, Santa Barbara, CA, USA. [2]Intel Labs, Hillsboro, OR, USA. ✉e-mail: kaustav@ece.ucsb.edu

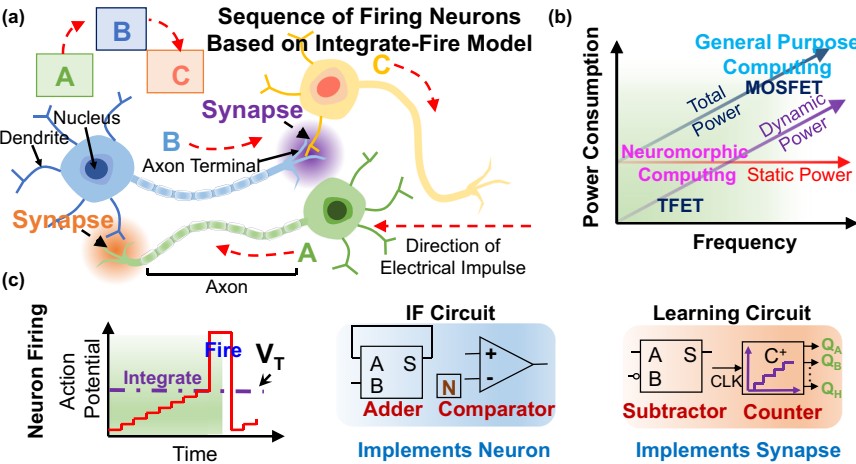

**Fig. 1 | Introduction to neuromorphic computing. a** Illustration showing the sequence of three firing neurons (A, B, C) in the brain, and their synaptic connections. The axon terminal of one neuron communicates with the dendrites of the other neuron through a small synaptic gap. **b** Operating power consumption of a circuit as a function of clock frequency. Dynamic power dissipation is lower for low-frequency neuromorphic circuits that mimic the neurons and synapses in the brain. **c** Diagram (left) illustrating the mechanism of a neuron firing in an integrate and fire (IF) model where buildup of its action potential past the firing threshold ($V_T$) causes it to fire. IF circuitry (center) implementing neuron firing is implemented by a feedback loop adder and comparator circuit (with a firing threshold of $N$), while neural learning (right) through synaptic behavior implementation is accomplished by a subtractor and a counter circuit ($C^+$ refers to an up-counter with Qs being the output). The neural learning circuit evaluates the synaptic weight between two firing neurons, allowing for the eventual training of the NM circuit.

theoretically be implemented with single devices capable of demonstrating both neuronal and synaptic behavior, the lack of any such practical implementations to date without the need for additional interface circuitry (degrades area- and energy-efficiency), makes implementations with digital-circuit-based approaches the most promising. In fact, some of the most successful commercial implementations of NM circuits to date, such as, Intel's Loihi[5] and IBM's TrueNorth[6] are digital-circuit-based approaches, realized with the ubiquitous complementary-metal-oxide-semiconductor (CMOS) technology. Furthermore, these digital circuits in scaled process nodes are suitable for high-volume manufacturing exploiting the inherent variation/noise-tolerance and easier programmability of digital circuits, as well as affordable monolithic integration of billions of neurosynaptic elements in a compact form factor. Additionally, digital implementations offer more flexibility and robust functionality at a system level that must integrate billions of such elements in compact form factors to reduce energy consumption related to signal transmissions across the elements. Although these digital implementations are therefore suitable for the implementation of dense, ultra energy-efficient, and high-performance NM systems, they are implemented with the ubiquitous CMOS technology, whose high OFF-state leakage current and degraded subthreshold swing ($SS \geq 60$ mV/decade; represents the ease with which the ON-current can be modulated with the application of gate voltage; more details in Supplementary Note 2, Supplementary Fig. 2) prevents further improvement in the energy-efficiency of the implemented NM circuit, and results in a best-case NM circuit energy consumption that is still 5 orders of magnitude larger than that of the human brain.

In this work, we show that the limited performance and energy efficiency of digital NM circuits implemented with CMOS can be circumvented by implementations with low OFF-state current and small-$SS$ (<60 mV/decade) tunneling field-effect-transistors (TFETs) (Supplementary Note 2)[10,11] that not only make them very promising for implementations of low-power circuits (Fig. 1b, c), but the low frequency of operation of NM circuits (~MHz) along with their low activity factors (because of the sparse firing activity of the neurons), makes TFETs highly desirable for these circuit applications. Particularly, TFETs designed with atomically thin two-dimensional (2D) materials[12–16] exhibiting pristine interface[12–17] and suppressed bandtails[18] offer excellent electrostatics and are relatively defect-free,

leading to steep turn-on characteristics[19–21] (average 4-decade $SS$ of 31[19] and 23[21] mV/decade) that are hard to achieve with conventional materials. In this regard, the judicious choice of a staggered source-channel heterojunction in well-designed lateral 2D-TFETs[20], which is also more large-scale manufacturable friendly w.r.t vertical 2D-TFETs[21–23], can simultaneously lead to large ON-current (>1 mA/µm) with small $SS$ (<60 mV/decade; can be as low as 20 mV/decade[20] over 4-decades of current swing), and is therefore, the best transistor choice for implementation of high-performance and low-power NM circuits. Therefore, the fully functional digital-NM circuit, along with its Hebbian learning circuitry introduced in this work, explores the benefits of employing 2D-TFETs in the NM computing space, where the digital nature of the circuit with its resilience to process variations, device-to-device variability, and suitability for implementing memory elements for storage of synaptic weights, also makes it very suitable for implementation of large-scale robust NM circuits. Comprehensive performance evaluations carried out w.r.t commercial 7 nm low-standby-power (LSTP) FinFET model[24] for various magnitudes of supply voltages ($V_{DD}$), activity factors ($AFs$), and under consideration of appropriate device and Cu-interconnect parasitics from the corresponding technology nodes[25], show that the 2D-TFET outperforms the CMOS at low $AFs$ by close to two orders of magnitude and represents a first such study of the application of 2D-TFETs in NM circuits. Besides bringing forth a novel application of TFETs in realistic scenarios where circuit performance is not hampered by their relatively lower ON-current and higher Miller capacitance, this study introduces an alternate hardware platform for designing high-performance, low-power robust NM circuits, thereby enabling the next generation of energy-efficient computing paradigm specifically targeted for applications such as AI.

## Results and discussion
The device design for implementing the NM circuit (Fig. 1c) is first optimized by tuning the device's physical parameters, and then its corresponding analytical model is implemented in Verilog-A (essentially a compact model that allows it to run smoother and faster without any convergence issues) for subsequent circuit simulation in HSPICE. The device design details and the circuit simulation methodology, including the criteria for analyzing the results, are discussed below.

## Device design

Lateral 2D-TFETs with a staggered 2D-transition metal dichalcogenide (TMD) WTe$_2$(source)-MoS$_2$(channel) heterojunction (i.e., 2D-HTJ-TFET)[20,26], simultaneously exhibit large ON-OFF current ratio with small *SS* characteristics, small leakage current, and a large ON-current that maximizes circuit performance, thereby enabling optimal NM circuit performance. Supplementary Fig. 3a in Supplementary Note 3 shows a double-gated (DG) variant of this device, where the DG architecture further enhances the device electrostatics, i.e., performance, and Supplementary Fig. 3b–f discusses the necessary physics needed for its accurate physics-based modeling. The device characteristics of the p- and n-2D-TFETs[26] are complementary due to equivalent electron- and hole masses and identical design parameters, thereby alleviating the need for sizing. Both the n- and p-2D-TFETs are designed to maximize source-channel electric field improving carrier injection, and minimize contact and access resistances. Moreover, both devices employ an intrinsic channel, a top- and bottom-gate dielectric thickness of 1 nm effective oxide thickness (EOT), and a channel length of 11 nm. The channel width has been chosen to yield the same device capacitance as that of a 7 nm FinFET device and is evaluated later.

The device capacitance, which is critical in regulating the circuit performance, is comprehensively modeled by accounting for both its intrinsic and parasitic components. The presence of a tunneling barrier at the source-channel junction of a TFET results in suppressed carrier injection w.r.t FinFETs, thereby leading to a smaller gate-source ($C_{GS}$) capacitance, while the larger channel charge near the channel-drain junction of the TFET leads to a larger gate-drain ($C_{GD}$) capacitance. Therefore, a 70-30 partition of the total 2D-TFET gate capacitance is allocated to $C_{GD}$ and $C_{GS}$[20,26], respectively.

## Circuit simulation and analysis

The analytical model of the 2D-TFET[26] is implemented in Verilog-A and subsequently employed to study the performance, robustness, and energy-efficiency of fundamental circuits employing 2D-TFETs: inverter, ring oscillator, and SRAM, leading into the development of the LIF NM circuit and its Hebbian learning circuitry. Operation of the NM circuit demonstrates digital neuron firing, which is analogous to the biological neuron firing in the human brain, along with the membrane potential leakage and learning behavior of two firing neurons (which is determined by their synaptic weight). Interconnect length of the corresponding Cu-interconnects was carefully chosen to achieve a capacitance equal to half of that of the total device capacitance, in accordance with conventional IC design practices, and therefore, to yield a realistic NM circuit performance. Performance projections were conducted by evaluating the static leakage ($E_{static}$) and dynamic switching ($E_{dynamic}$) energies at various $V_{DD}$ and *AF* ranging from 0.2 V to 0.7 V and 1 to $10^{-6}$, respectively, from which the total dissipated energy ($E_{total}$) is evaluated. The $E_{total}$ is compared against implementations of NM circuit with LSTP FinFET to evaluate the magnitude of energy savings possible with 2D-TFET implementations.

This section analyzes the design, performance, robustness, and energy-efficiency of 2D-TFETs in fundamental circuits comprising inverters, ring oscillators, and SRAMs, necessary for the development of the LIF NM circuit as shown later in the "NM circuit" subsection.

## Inverter implementation

The static and dynamic performance of 2D-TFETs in an inverter circuit driving a particular output capacitance is studied in this subsection, which seamlessly lends to the understanding of more complex logic gates and circuits. However, the very performance of the basic inverters is determined by the transfer characteristics of the individual transistors (TFET and LSTP FinFET), which are described in Supplementary Fig. 4 in Supplementary Note 4. Figure 2 shows the

simulation of the static (Fig. 2a) and dynamic (Fig. 2b) characteristics of a unit-sized inverter, simulated by sweeping the input voltage ($V_{IN}$) and plotting the corresponding output voltage ($V_{OUT}$). As seen from Fig. 2a, although the transfer characteristics of both 2D-TFET and the 7 nm FinFET transistors are quite similar, the 2D-TFET inverter demonstrates a slightly higher peak gain of ~18 compared to that of ~12 in the latter, resulting from the better saturation of the drain current in the former. This higher gain also leads to reduced short-circuit and dynamic power consumption, as well as better noise margins.

The dynamic characteristics of the inverters have been simulated by varying the inverter output (load) capacitance ($C_{OUT}$), connected to the inverter output across a 1000 nm long interconnect (assumed for simulations of the NM circuit, described in "NM circuit" subsection), from 1 aF to 1 fF. By evaluating the delay ($t_p$) of the input-to-output transition, and the instantaneous current drawn from the supply during this transition, the average power dissipation, and the energy-delay-product (*EDP*), is evaluated for both the 2D-TFET and the FinFET implementations. The higher delay of the 2D-TFET (due to its lower ON-current) translates to higher EDP, and the EDP metrics get worse as the load capacitance is further increased. In fact, as will be shown later, the main advantages of TFETs are in implementations of sparse switching circuits where its much lower OFF-current and small *SS* help in lowering the static power dissipation, thereby improving the overall performance.

## Ring oscillator

Figure 2c shows an 11-stage ring oscillator, implemented considering both interconnect and device parasitics, and designed with minimum sized 2D-TFET and FinFET inverters. Figure 2d, e compares the transient characteristics of the FinFET and the 2D-TFET ring oscillators, from which the frequency of oscillation is extracted to be 10 GHz and 57 MHz, respectively, corresponding to single-stage delays of 10 ps and 1.6 ns. The delay of the 2D-TFET ring oscillator is larger due to its lower ON-current. The effect of the enhanced Miller capacitance in creating large overshoots and undershoots of the output voltage in TFETs is also observed in Fig. 2e.

## SRAM design

Static random-access memory (SRAMs), which occupy up to 70% of the processor area are the main memory elements in designing CPU cache memory offering fast memory access and can be used for synapse weight retention in a designed NM system comprising of several neurons. However, this large prevalence of SRAMs also results in a large power consumption. In fact, SRAM data access in Intel's Loihi[5] has been estimated to be more energy intensive than each neuronal spike, necessitating the development of low-power SRAM implementations. Although SRAM design with 2D-TFETs can improve the energy-efficiency, the standard SRAM design utilizes two access transistors for operation, which require bidirectional current flow, and are therefore, ill-suited for implementation with unidirectional-TFETs. This necessitates the development of a modified SRAM design, which either uses a pass transistor network of TFETs, or solitary 2D-FETs, for implementing the function of the access transistors (Fig. 2f–l).

Figure 2f–h are SRAM designs implemented with 2D-FET access transistors[27] with a $V_{DD}$ of 0.9 V. Non-ideal effects of source-drain series resistance and mobility degradation are considered along with presence of interface traps for accurately modeling a realistic 2D-FET device performance. Bit-line capacitances considering associated wire capacitances of both *BL* and $\overline{BL}$ are assumed to be present, equivalent to 25 fF. Similarly, Fig. 2i, j are SRAM designs implemented with a pass transistor network of 2D-TFETs for the access transistors, and Fig. 2k, l demonstrate the noise margins and the read operation of the all-2D-TFET SRAM design. The channel width for all the 2D-TFETs has been

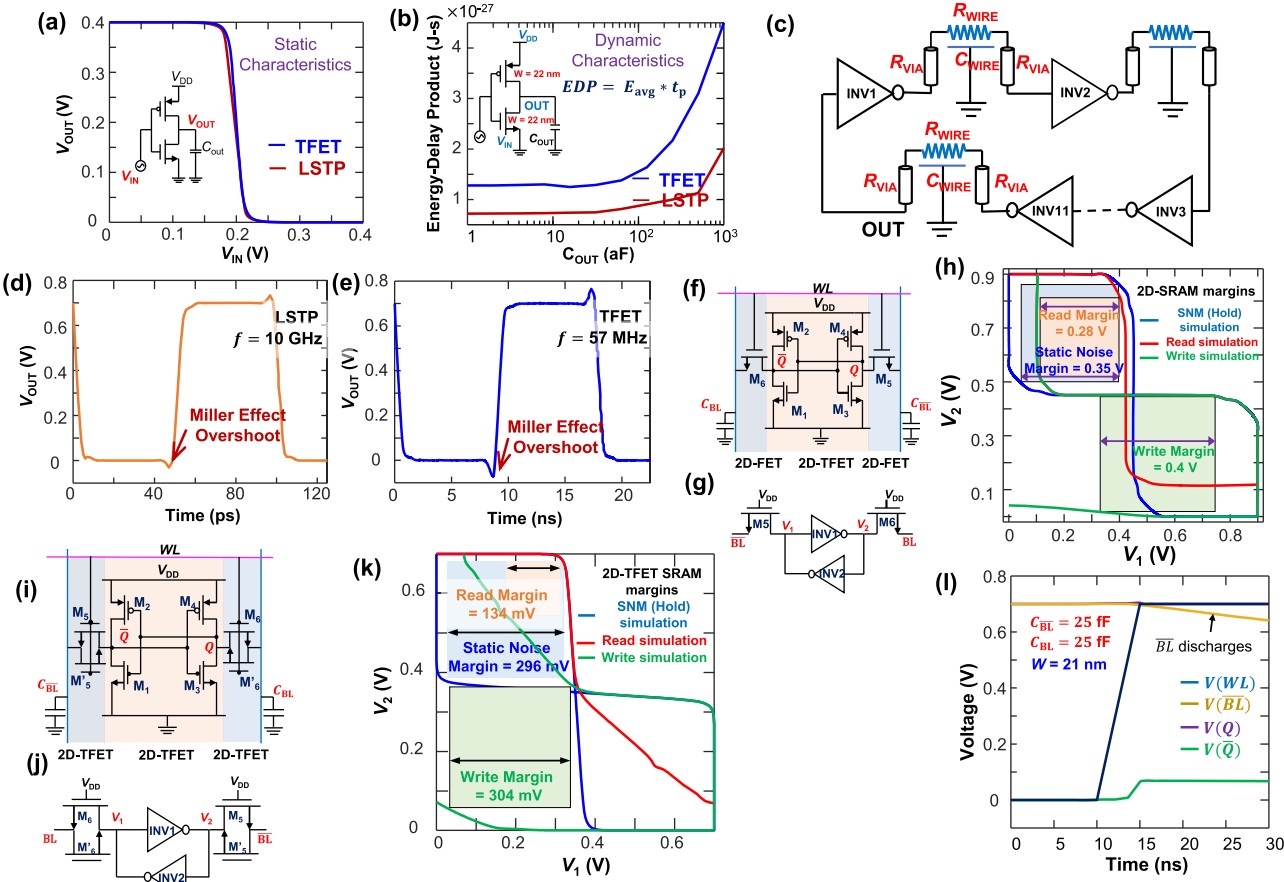

**Fig. 2 | Design and implementation of standard logic gates. a** Static characteristics of a minimum sized 2D-TFET- and LSTP- inverter, implemented with transistors of the same effective width and channel length, and simulated at $V_{DD} = 0.4$ V. **b** Energy-Delay-Product (*EDP*) comparison of a 2D-TFET- and LSTP-transistor driving an output capacitance varying from 1 aF to 1 fF. The delay ($t_p = t_2 - t_1$) has been calculated from the time taken by the input signal to reach $V_{DD}/2$ ($t_1$) to the time taken for voltage at the output node (OUT) to drop to $V_{DD}/2$ ($t_2$). The average energy consumption is calculated by multiplying the average power consumed during $t_1$ to $t_2$ with the transition time ($t_p$). **c** Schematic of the 11-stage ring oscillator considering interconnect parasitics and minimum-sized inverters. Ring oscillator implemented with minimum sized **d** LSTP and **e** 2D-TFET with an oscillation frequency of 10 GHz and 57 MHz, respectively. Enhanced Miller overshoot is observed in **e**. **f** All-2D implementation of a 6T-SRAM design, with 2D-FETs being the access transistors and 2D-TFETs being the inverters. Schematic shows the Word Line (*WL*) and Bit Line (*BL*) signals. **g** Simplified schematic of **f** showing probed node voltages $V_1$ and $V_2$. **h** Noise margins of the simulated SRAM circuit at $V_{DD} = 0.9$ V. The static noise margin (hold margin) has been simulated by sweeping $V_1$ and $V_2$, keeping M5 and M6 disconnected, while for read (Q=1) and write (Q=1) margin simulations, $BL$ and $\overline{BL}$ have been connected to $V_{DD}/V_{DD}$ and 0/ $V_{DD}$, respectively. Read margin is the lowest at 0.28 V. **i** All-2D-TFET-SRAM designed completely with 2D-TFETs, which addresses the problem of unidirectionality in TFET current transport. **j** Simplified sketch of **i**. **k** Hold, read (Q=1), and write (Q=0) margin simulations of the all-2D-TFET SRAM simulated with a $V_{DD}$ of 0.7 V assuming the same bias conditions for the Bit Lines as in **h**. The noise margins are similar as compared to those in **h** even with a reduced supply voltage of 0.7 V w.r.t 0.9 V in **h**. **l** Simulation of node voltages during read operation of the SRAM in **i**. Plot shows a small increase of the node voltage of $\overline{Q}$ and a slight decrease of the $\overline{BL}$ voltage, but below the inverter tripping voltage.

assumed to be 21.25 nm in accordance with the device width chosen for the NM circuit simulation, along with a channel length of 11 nm. INV1 in Fig. 2g, j corresponds to transistors $M_1$ and $M_2$, while INV2 corresponds to transistors $M_3$ and $M_4$. The output of the first inverter (INV1) is $V_2$ while for INV2 it is $V_1$.

As seen from Fig. 2h, the sharp transfer characteristics of the 2D-TFET inverters with 2D-FET access transistors leads to an impressive static noise margin (SNM) (or hold margin) of 0.35 V in a 0.9 V $V_{DD}$ operation, which demonstrates the ruggedness of the SRAM during data retention. The write margin is highest at 0.4 V, while the read margin is 0.28 V. Likewise, the read, write and hold margins for the all-2D-TFET-SRAM cell, simulated with a $V_{DD}$ of 0.7 V, are 133 mV, 304 mV and 296 mV, respectively. The obtained noise margins are excellent, thereby proving the merit of using 2D-TFET pass-transistor network for the access transistors. Having demonstrated the performance of 2D-TFETs in basic inverter circuits and SRAMs, the next subsection discusses the design of the NM circuit which utilizes the circuit components discussed till now.

## NM circuit

This subsection introduces and analyzes the performance of the fully functional digital LIF-NM circuit, implemented with both 2D-TFETs and LSTP-FinFETs. A sequence of digital firing neurons, each emulating a neuron in the human brain, are designed such that the membrane potential of each neuron increases with every clock cycle (typically employed in a digital circuit to synchronize circuit operation) during the 'integrate' operation, and subsequently decays during the 'leakage' operation. These neurons are separated from each other by digital synapses—akin to the neuronal connections in the brain, and are responsible for the learning behavior (Fig. 1a). While a higher synaptic weight refers to stronger correlation between two neurons, a low synaptic weight refers to uncorrelated neurons. In this work, the synaptic weight varies from a minimum of -1 (uncorrelated) to a maximum of 1 (correlated), with a synaptic of 0 referring to disconnected neurons. This synaptic weight update is governed by the spike time-dependent learning plasticity (STDP) rule and is implemented with Hebbian learning style[28], which increases/decreases the

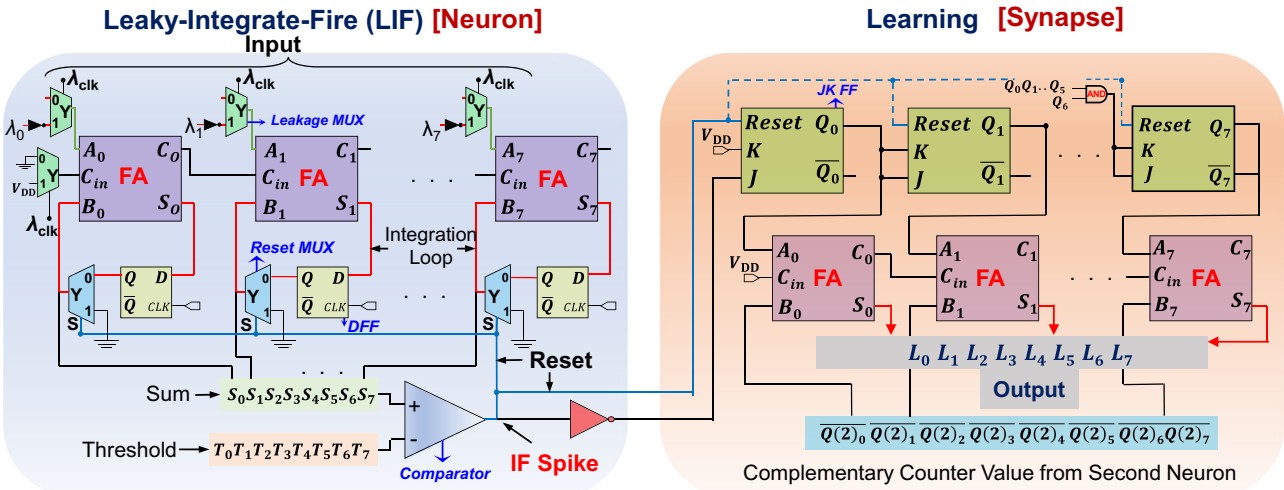

**Fig. 3 | Designed neuromorphic circuit.** Entire schematic of the designed neuromorphic circuit architecture for one firing neuron, containing both its Leaky-Integrate-Fire (LIF) (left) and Hebbian learning circuitry (right). In the circuit block to the left, "IF Spike" denotes the node where the neuron firing is observed. The green MUX's at the top of the LIF circuit (Leakage MUX) are responsible for implementing the leakage operation of the membrane potential ($S_0S_1...S_7$) once the leakage clock ($\lambda_{clk}$) is asserted, subtracting the leakage potential ($\lambda_0\lambda_1..\lambda_7$) from the membrane potential ($S_0S_1..S_7$) through the addition of the 2's complement of the leakage potential to the membrane potential. The full-adders (FAs) in the circuit are indicated in purple, implementing the integrate operation of the membrane potential through an integration loop, going through the D-Flip Flops (DFFs) and the 2:1 MUXs in blue (also responsible for resetting the membrane potential after successful neuron fire−Reset MUX). The comparator responsible for the neuronal spike is shown at the bottom end of the figure, which generates the IF (integrate fire) spike once the membrane potential exceeds the preset threshold ($T_0T_1..T_7$).

The circuit block on the right implements the Hebbian learning part of the circuitry, where an up-counter is designed based on JK Flip-Flops (JK FF) that resets every time the neuron spikes, thereby keeping track of the time elapsed since the last neuronal spike. The counter value of the up-counter of the neuron is compared against the complementary of the counter value of the secondary neuron ($Q(2)_0Q(2)_1..Q(2)_7$) through the adder circuit indicated in purple, generating a signed output of $L_0L_1..L_7$, which indicates both the causality (the sign bit) and the time difference of the firing events of the primary and secondary neurons. This time difference indicates the strength of the neuronal connection of the two neurons, with a smaller time difference indicating a smaller time elapsed between the two firing events, thereby indicating a strong neuronal connection. The counter circuit is implemented by the JK FF, and the subtractor circuit is implemented with the FA circuitry. For the second JK FF (which implements the output of $Q_1$), the inputs to the JK terminals are derived from $Q_0$. For all subsequent JK FFs, the inputs are generated by taking the AND product of the outputs from the preceding JK FFs.

---

synaptic weight for causal/non-causal firing events (causal/non-causal firing events refer to the pre-synaptic neuron firing before/after the post-synaptic neuron, respectively), with the magnitude of the weight update being determined by the time difference between the two firing events. The implemented neuron membrane potential has been assumed to be of 8-bit resolution, thereby offering a good compromise between the simulation runtime and accuracy.

**Neuromorphic circuit implementation**
Figure. 3 shows the implemented NM circuit (corresponding to Fig. 1c) with its associated LIF neuron and the learning circuitry (synapse), involving a total of around 3259 transistors and more than $10^5$ nets. The LIF logic comprises an 8-bit full adder (FA) (see details on implementation of a digital FA in Supplementary Note 5), whose output represents the membrane potential of the neuron. The FA (Supplementary Figs. 5 and 6) has been configured such that it implements the integrate function, with the integration loop going through a network of D-flip flops (DFFs) and 2:1 MUXs (indicated in blue). While the DFFs are chosen to retain the output of the full-adders from the previous clock cycle, the MUXs are used to implement the reset operation through their select line, which is the output of an 8-bit comparator and is responsible for the spiking operation of the neuron. When the select line ($S$) of a blue MUX is low, the adder output ($S_0S_1..S_7$) from the last clock cycle feeds back into the input terminal of the adder (inputs $B_0B_1..B_7$ shown with red lines), thereby implementing the integrate functionality and increasing the neuron membrane potential. For high value of the select line, however, the output of the MUX is shorted to its input terminal of 1, which is grounded, thereby resetting the output of the adder, and hence, the neuron membrane potential. Although the circuit in this work employs an 8-bit FA leading to an 8-bit membrane potential, it can easily be extended to accommodate more bits if

higher accuracy is desired. The leakage operation of the NM circuit is implemented with the 'Leakage MUXs' in green (Fig. 3) with their outputs feeding into both the input terminals of the 8-bit FAs ($A_0A_1...A_7$) and carry terminal ($C_{in}$) of the input FA. These leakage MUXs either accept inputs from the NM array (consisting of numerous circuit blocks shown in Fig. 3 whose output is connected to input 0 of these MUXs) for the low phase of the leakage clock ($\lambda_{clk}=0$) leading to 'integrate' operation (through the integration loop described earlier), or the 1's complement (i.e., complement of a binary number; more details in Supplementary Note 6) of the membrane leakage signal ($\lambda_0\lambda_1..\lambda_7$, connected to input 1 of these MUXs) during the leakage clock's high phase ($\lambda_{clk}=1$) leading to 'leakage' operation. This leakage operation, i.e., subtraction of the leakage potential from the membrane potential, is in fact, accomplished by adding the membrane potential to the 1's complement of the leakage potential along with an additional input of binary 1 (accomplished through the input carry bit of 1 to the FA during $\lambda_{clk}=1$), i.e., adding the membrane potential to the 2's complement of the leakage potential, as described in detail in Supplementary Note 6. The resulting output of the FA circuit ($S_0S_1..S_7$) during the leakage operation therefore, decreases by applied membrane leakage potential every clock cycle for the duration during which $\lambda_{clk}$ is high, thereby mimicking the neuronal membrane leakage operation.

The neuron firing and membrane potential reset operation in the circuit is accomplished by the 8-bit comparator implemented at the bottom of the LIF circuit, which compares the membrane potential ($S_0S_1...S_7$) w.r.t the threshold (bits $T_0T_1...T_7$, more details in the following subsection) and generates an output signal (goes high to $V_{CC}$) when the membrane potential exceeds the threshold. This phenomenon of the output of the comparator going high is akin to the firing of the neuron, and it simultaneously causes the output of the blue MUXs

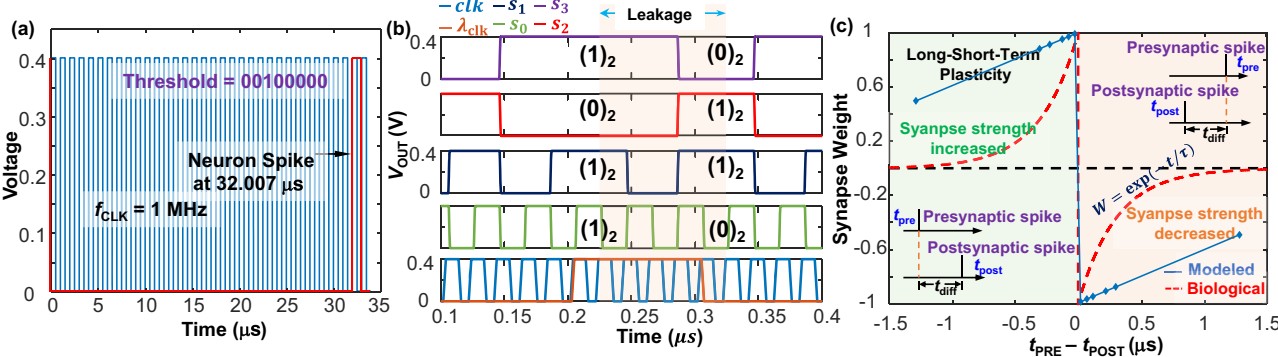

**Fig. 4 | Demonstration of neuromorphic circuit operation. a** Simulation of the neuron spiking event at the output of the comparator in Fig. 3. Circuit has been simulated under a clock frequency of 1 MHz, an input of decimal 1 to the integrate array and a comparator threshold of $(32)_{10}$. Neuron spike happens at 32 μs (corresponding to the clock frequency and waveform shown in bright red) after which the circuit resets. **b** Shows the leakage operation of the membrane potential. The assertion of the leakage clock ($\lambda_{clk}$) for 5 clock periods for a leakage potential $(\lambda_0\lambda_1..\lambda_7)$ of $(1)_{10}$ lead to the decay of the membrane potential from $(1011)_2$ (before assertion of $\lambda_{clk}$) to $(0110)_2$ (after assertion of $\lambda_{clk}$), after which normal buildup of the membrane potential continues. The clock frequency has been assumed to be 50 MHz during the simulation. The decay in the membrane potential from $(11)_{10}$ to $(6)_{10}$, i.e., a decrease of $(5)_{10}$ corresponds to a decrease of $(1)_{10}$ every clock cycle for the 5 clock cycles that $\lambda_{clk}$ is asserted for, and is due to the application of a leakage potential of $(1)_{10}$. The one clock delay in the leakage operation once the leakage clock is asserted is due to the state retention in the DFFs of the integrator loop. Also, the leakage operation is only shown for bits $S_0$ to $S_3$ (of the FA in the integration loop) since the higher order bits do not change. **c** Long short-term plasticity simulation (solid blue line) of the neural synapse and its comparison with the biological model (dashed red line). The neuron firing time difference ($t_{DIFF}$) has

been calculated by assuming a constant firing event for the secondary neuron (post-synaptic neuron, firing event at $t = t_{POST}$) while varying the firing events of the primary neuron (pre-synaptic neuron, firing event at $t = t_{PRE}$). The firing of the pre-synaptic neuron has been assumed to drive the firing event of the post-synaptic neuron. Simulation assumes that the up-counter in the learning circuitry for the neurons counts up by $(1)_{10}$ every clock cycle. For a causal firing event where the post-synaptic neuron fires after the firing of the pre-synaptic neuron (i.e., $t_{DIFF}$ is negative), the synaptic weight is positive and it increases (to a maximum value of 1) as $t_{DIFF}$ is reduced, i.e., when the timing difference between the firing events of the pre- and post-synaptic neurons reduces, thereby, implying a strong synaptic connection. Similarly, for an anti-causal firing event where the post-synaptic neuron fires before the firing of the pre-synaptic neuron (i.e., $t_{DIFF}$ is positive), the synaptic weight reduces (minimum value is −1) as $t_{DIFF}$ is reduced, thereby implying that the post-synaptic neuron drives the firing events of the pre-synaptic neuron. Sharpness of STDP curve depends on the increase in the counter value per clock cycle. Also, shown is the biological STDP curve, whose synaptic weight ($W$) is exponential with time ($t$), and is a function of the response time $\tau$ (regulates the sharpness of the synaptic weight decay).

in the LIF circuit to be grounded (the output of the comparator is the select line for the MUXs; therefore, for high value of the select line, the output of the MUX is shorted to the input terminal 1 of the MUX, which is in fact grounded). Furthermore, since the threshold for the comparator firing can be pre-programmed, it allows for fine-tuning the firing threshold of each individual firing neuron separately, which provides an additional knob for the synaptic weights to be tuned, thereby increasing the training/learning efficiency of the circuit. Additionally, to maximize the circuit operating frequency and to ensure the correct circuit operation, it becomes imperative that the comparator completes the computation (comparing the membrane potential against the threshold) and resets the membrane potential before the next clock pulse arrives at the FA and the membrane potential is updated, thereby making the comparator along with the integration loop the critical path of the circuit. Therefore, optimum circuit operation is ensured by designing the comparator to deliver the least computation delay, and is achieved such that it starts by comparing the most significant bits (MSB) of the two signals (membrane potential and threshold; if the MSB of the membrane potential is already larger than that of the threshold then the comparator readily fires and there is no need to compare the other bits) and progresses down to the least significant bit if all the higher bits are equal.

The output of the comparator is also connected to the learning circuitry where it resets the 'up counter' (implemented with the reset terminal in the JK flip-flops of the learning circuitry of Fig. 3; more details on implementation in Supplementary Note 5B, Supplementary Fig. 7), which counts up by $(1)_{10}$ every clock cycle and resetting every time the neuron fires (i.e., the Reset signal of the JK FFs is asserted). Therefore, the up-counter is responsible for counting the number of clock cycles elapsed since the last neuron spike. For evaluating the operation of the learning circuit and simulating the STDP behavior of two firing neurons, the counter value of a secondary neuron is also

assumed $(Q(2)_0Q(2)_1..Q(2)_7$, shown on the bottom right of Fig. 3; the LIF and learning circuitry for the secondary neuron is not simulated for simplicity). While the firing event of the primary neuron (the circuit on the left shown in Fig. 3) is varied by either changing the input to the integrator array (through the input terminal 0 of the green MUXs feeding into the FAs of the LIF circuitry), or by modulating the comparator threshold, a constant firing event is assumed for the secondary neuron. Therefore, this allows for simulating various time differences between the firing events of the two neurons, necessary for simulation of the eventual STDP behavior. This time difference in the firing events is, in fact, obtained by subtracting the counter value (i.e., the output of the JK FF array) of the secondary neuron from the counter value of the primary neuron $(Q_0Q_1..Q_7)$, and is accomplished by adding the 2's complement of the counter value of the secondary neuron to the counter value of the primary neuron employing the FA (shown in purple) of the learning circuitry. This operation is very similar to the leakage operation described earlier for the LIF circuit, and more details on the subtraction operation are provided in Supplementary Note 6. Finally, the resulting sign-bit of the subtraction operation (bit $C_7$) determines the polarity of the result, i.e., a non-zero value of $C_7$ implies that the difference of the counter values of the primary and secondary neurons, i.e., output $(L_0L_1..L_7)$, is positive, while a zero value of $C_7$ implies that the output is negative. Finally, the output $L_0L_1..L_7$ is normalized to the maximum value of a 8-bit number, and the STDP curves are obtained, as described in detail in the following subsection.

### Spike time-dependent plasticity and output spike simulation
Figure 4 shows the simulation of the output spike and the STDP behavior of two firing neurons (the primary neuron in its entirety with its learning circuitry is shown in Fig. 3, while only the counter value of the secondary neuron is shown on the bottom right of Fig. 3), simulated for various instances of the primary neuron spiking. A clock frequency

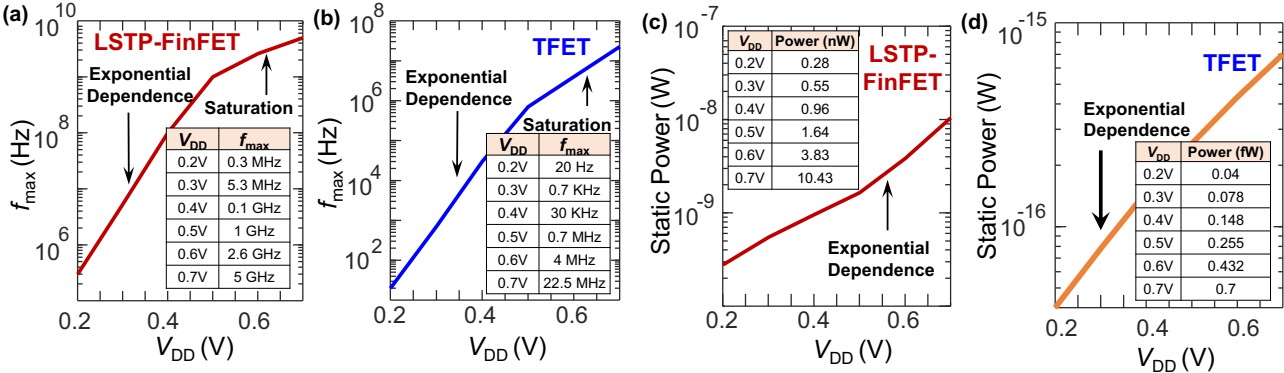

**Fig. 5 | Performance analysis—I.** Simulated $f_{max}$ vs $V_{DD}$ for **a** LSTP-FinFET and **b** TFET implementations. Inset table shows the frequencies corresponding to the simulated $V_{DD}$. The static power consumption of the NM circuit simulated with **c** LSTP-FinFET and **d** TFET at $V_{DD}$ ranging from 0.2 V to 0.7 V. Inset table shows the

static power dissipation corresponding to the simulated $V_{DD}$. While the $f_{max}$ of the TFET is lower than that of the LSTP-FinFET, its static power dissipation is orders of magnitude lower, thereby improving the overall performance.

of 1 MHz has been assumed for the simulations, with a membrane threshold of $(32)_{10}$ and an input of $(1)_{10}$ to the integrator array of the primary neuron. The neuron spike under these conditions thus happens at around 32 clock cycles, i.e., at $t = 32\,\mu s$ (Fig. 4a) when the neuron membrane potential equals the threshold (Fig. 1c) and the leakage operation is absent (leakage clock is not asserted, i.e., $\lambda_{clk} = 0$). This neuron spiking event can, however, be hastened by using a faster clock, having a lower firing (comparator) threshold, or by having an input of higher magnitude (from the NM array through the input terminal 0 of the green MUXs in LIF circuit) to the neuron integrator array. Similarly, Fig. 4b shows the leakage operation ($\lambda_{clk} = 1$) of the neuron where the membrane potential decreases from $(11)_{10}$ to $(6)_{10}$ upon the application of a leakage potential ($\lambda_7\lambda_6..\lambda_0$) of $(1)_{10}$ for five clock cycles. Once the neuron, i.e., comparator of the LIF circuit spikes, the Reset signal (Fig. 3) is generated at the output of the comparator which resets the membrane potential, i.e., sets the output of the 8-bit FA to zero. This reset signal is also fed to the reset input of the up-counter array of the learning circuit (implemented with JK FFs) for accomplishing the reset operation of the counter value of the primary neuron every time the neuron fires, while the inverted output of the comparator (implemented through the red inverter of the LIF circuit, immediately following the comparator) ensures correct operation of the up-counter. The output of this up-counter ($Q_7Q_6..Q_0$) is fed to the first input of a FA in the learning circuit (denoted in purple in Fig. 3), while the 1's complement of the counter value of the secondary neuron $\left(Q(2)_7Q(2)_6..Q(2)_0\right)$ is connected to its second input, which along with the input carry of binary 1 to the FA in the learning circuitry evaluates the difference in the counter values of the two neurons (see Supplementary Note 6), i.e., evaluates the timing difference between the two firing events (since the counter value resets every neuronal spike, hence, the counter value at a particular time measures the time elapsed since the last neuronal spike) based on the mechanism described in the preceding subsection.

This subtraction operation generates a carry bit (at $C_7$) which acts as a sign bit and determines the causality of these two firing events (Supplementary Fig. 8) (firing events for the neuron and the synapse), i.e., for a zero-sign bit ($C_7 = 0_2$) the difference is negative, which means that the counter value of the secondary neuron is higher than that of the primary neuron. Since this happens when the secondary (post-synaptic neuron, since it is assumed that the primary neuron drives the firing events of the secondary neuron) neuron fires before the spiking event of the primary (pre-synaptic) neuron, this refers to an anti-causal relationship between these two firing events. Similarly, when the firing event of the pre-synaptic neuron happens earlier than that of the post-synaptic neuron, the counter value of the primary neuron is larger than

the counter value of the secondary neuron and their difference ($L_0L_1..L_7$) is positive, and therefore, the MSB of the subtractor is non-zero ($C_7 = 1_2$).

The learning curve simulated in Fig. 4c for two firing neurons, shows the change of their synaptic weight as the time difference between their spiking events is varied. Since the output of the subtractor ($L_0L_1..L_7$) implemented employing the FA in the learning circuit is a 8-bit output, disregarding the sign bit at $C_7$, it has been normalized w.r.t to the highest 8-bit number $(11111111)_2$, $(511)_{10}$, to yield a synaptic weight varying between 1 (causal response) and -1 (anti-causal response). The learning curves for both positive and negative $t_{DIFF}$ (defined as $t_{DIFF} = t_{PRE} - t_{POST}$, $t_{PRE}$ and $t_{POST}$ are firing events for the pre- and post-synaptic neurons, respectively) are simulated by changing the timing events for the pre-synaptic (primary) neuron. Although the simulated STDP learning curve corresponds well with that of the biological counterpart as shown in Fig. 4c, our use of the subtractor circuits for STDP simulation only allows for a linear behavior of the synaptic weight update with $t_{DIFF}$. Use of more complex circuits can introduce higher order polynomials into the behavior, and it may approach the biological exponential behavior. Important to note however, is that the steepness of the STDP response (rate of change of synaptic weight with $t_{DIFF}$) can be increased by increasing the number of counts that the up-counter counts per clock cycle, which has been assumed to be 1 in our simulations.

## Maximum frequency of operation

The performance of any NM circuit is best given by the energy dissipated per neuron spike [5,6], which depends on the circuit capacitance, static/dynamic power dissipation, and the maximum frequency of operation ($f_{max}$) of the circuit at a particular $V_{DD}$. The $f_{max}$ is determined by the net circuit capacitance and the ON-current of the transistors, with a smaller capacitance and a larger ON-current resulting in a larger $f_{max}$ and a correspondingly smaller energy dissipation, assuming a constant static power dissipation (at a particular $V_{DD}$). Extraction of the $f_{max}$ of the NM circuit with both TFET and FinFET implementations at varying $V_{DD}$ ranging from 0.2 V to 0.7 V in steps of 0.1 V in Fig. 5a, b shows an initial exponential increase (due to exponential dependence of the ON-current with $V_{DD}$ in the subthreshold regime, Supplementary Fig. 4) until it starts saturating at higher $V_{DD}$ when the increase in the ON-current with $V_{DD}$ gets roughly linear. Additionally, although the $f_{max}$ of the TFET at a particular $V_{DD}$ is smaller w.r.t the FinFET due to the limited ON-current of the former (Supplementary Fig. 4), the smaller *SS* of the TFET results in a faster increase of $f_{max}$ with $V_{DD}$ (Fig. 5b), thereby allowing the TFET NM implementation to span a larger frequency range of operation of around 6 decades, compared to that of 4 decades for the FinFET.

However, for an accurate measure of the $f_{max}$, the net circuit capacitances−including both interconnect and device (both parasitic and intrinsic), of the two implementations must be made equal. While the equivalence of the device capacitances is ensured by varying the width of the TFET to result in an equal circuit capacitance with the FinFET (without the presence of interconnect capacitance), the interconnect length connecting each logic gate (assumed to be same for connections between all logic gates in both the implementations) is varied to obtain a total interconnect capacitance equal to half of the net device capacitance, in accordance with conventional design practices. The device (Supplementary Fig. 9a, b in Supplementary Note 7)- and the interconnect (Supplementary Fig. 9c,d)-capacitances for both the TFET and FinFET implementations (Supplementary Fig. 10) are extracted from the slope of the power-frequency curve at a particular $V_{DD}$. Additionally, although the parasitic sidewall body-source/drain capacitances present in thicker body FinFET devices (Supplementary Fig. 9) are absent in the intrinsically thin-body 2D-TFET device, these capacitances are set to zero in the FinFET NM circuit for their best-case circuit performance estimation. An average device capacitance of 46.4 fF for both TFET and FinFET implementations is extracted (Supplementary Fig. 10, Supplementary Fig. 11), resulting in an interconnect length of 1000 nm (between each logic block), for a net interconnect capacitance of 23 fF (half of that of the device capacitance), and hence, a total circuit capacitance of ~70 fF (Supplementary Fig. 10b). Further details on the device and interconnect capacitance models are provided in Supplementary Note 7.

### Calculating the static power dissipation

Since NM circuits are, in general, very sparse firing circuits where a spike can happen as infrequently as with a frequency of several Hz, the most dominant power dissipation mechanism in a NM circuit is the static power dissipation component. Therefore, devices with a lower OFF-current, i.e., TFETs (Supplementary Fig. 2,4) whose carrier distribution in the source is suppressed due to Boltzmann tail-cutoff[11], demonstrate an overall lower static power consumption of the entire circuit. This benefit in energy-efficiency for TFETs is even more significant in circuits with lower $AF$s, since a lower $AF$ makes the static power component more significant w.r.t the dynamic power component, therefore determining the overall energy efficiency of these circuits. Figure 5c, d show the static power consumption of the NM circuit with both 2D-TFET and FinFET implementations, simulated at $V_{DD}$ varying from 0.2 V to 0.7 V. As seen from Fig. 5c, d, although the static power increases exponentially with $V_{DD}$ for both 2D-TFET and FinFET implementations, the static power consumption of the 2D-TFET circuit is still orders of magnitude lower than that for the FinFET across the entire $V_{DD}$ range. However, a lower static power dissipation only lowers the overall energy dissipation if the corresponding $f_{max}$ of the circuit is comparatively higher, since such a circuit dissipates the static power over a shorter time duration, thereby dissipating lower static energy overall (simulations assume a duty factor of 0.5; duty factor is defined as the $t_{ON}/T$, where $t_{ON}$ is the time for which the signal is high and $T$ is the total time period). The net energy dissipation, which depends on the static power dissipation and maximum frequency of operation, which in turn depends on the net circuit capacitance, supply voltage, and the activity factor, is described in detail in the next subsection.

### Energy efficiency of 2D-TFET vs FinFET implementation

As already stated, the energy dissipation per neuron spike is the most important metric determining the overall efficacy of the circuit implementation. This net energy dissipation ($E_{total}$) in a circuit per clock cycle has two components−$E_{static}$ and $E_{dynamic}$, where the latter depends on the $AF$ of the circuit. If the static power dissipation of the circuit at a particular $V_{DD}$ is $P_{static}$, the entire circuit capacitance is $C_L$, and the frequency is $f$, then the total energy dissipation can be expressed as:

$$E_{total} = (P_{static}/f) + AF.C_L V_{DD}^2 \qquad (1)$$

From this expression, it is evident that the least energy dissipation in the circuit happens when the frequency of operation is the highest ($f_{max}$), hence, for all eventual simulations, the energy dissipation at this highest frequency of operation (Fig. 5) is evaluated. Additionally, the energy dissipation is evaluated at $AF$s ranging from 1 to $10^{-6}$ in the sparse firing NM circuits. Also, as seen from (1), although the dynamic energy dissipation decreases with a decrease in $V_{DD}$, however, both $P_{static}$ and $f_{max}$ also decrease exponentially (Fig. 5). If the decrease in $f_{max}$ is steeper than that of $P_{static}$, the total energy dissipation in the circuit will increase at lower $V_{DD}$. However, this is only important at very low $AF$s when the static energy component is larger than the dynamic energy dissipation. Therefore, to find the ideal $V_{DD}$, and hence, the ideal frequency of operation of the circuit, the energy dissipation of the circuit at various $V_{DD}$ ranging from 0.2 V to 0.7 V and at various $AF$s (ranging from 1 to $10^{-6}$) at corresponding $f_{max}$ needs to be evaluated, which is shown in Supplementary Fig. 12 (Supplementary Note 8) for the FinFET, and in Supplementary Fig. 13 for the 2D-TFET implementations. Also, as is apparent from Supplementary Figs. 12 and 13, the energy dissipation−$V_{DD}$ curve has a minimum at a particular $V_{DD}$ (depending on the activity factor), which therefore, is the most optimum biasing condition for the circuit to work. Hence, Supplementary Fig. 14 (Supplementary Note 9) compares the energy consumption of both the 2D-TFET and the FinFET models at a particular $AF$, and at their respective optimal $V_{DD}$ of operation. Figure 6a compares the energy dissipation at the lowest $AF$ where the benefits of the TFET implementation are highest w.r.t to the FinFET implementation, while Fig. 6b and Fig. 6c compares the energy dissipation as a function of $V_{DD}$ (at corresponding $f_{max}$) and activity factor, respectively. Please note that while Fig. 6a computes the energy consumption of the circuit per clock cycle for both the TFET and the FinFET implementations, the total energy consumption per neuron spike for both these implementations can be computed by multiplying the corresponding energy dissipation metric with the total number of clock cycles elapsed between two successive neuron spikes, which is 32 in our case.

As seen from Fig. 6a, the energy dissipation of the 2D-TFET implementation is lower than that of the FinFET circuit across the entire frequency range for $AF$ of $10^{-6}$, due to the total dissipated energy being limited by the orders of magnitude lower static energy of the TFET at this lowest $AF$. Also, interesting to note is that the minimum energy dissipation is at two different frequencies (and corresponding $V_{DD}$) for the TFET and the FinFET, and happens due to the tradeoffs between $f_{max}$ and $P_{static}$ for the two implementations. Particularly, as the dynamic energy consumption remains independent of the operational frequency while the static energy is inversely related to it, an increase in $f_{max}$ leads to a decrease in the overall computational energy. This holds true unless the increase in $f_{max}$ is accompanied by a corresponding increase in $V_{DD}$, causing both static and dynamic energies to increase linearly and quadratically, respectively, which is why the minimum computational energy is obtained at an intermediate operational frequency. Figure 6b shows the ratio of the energy dissipation of the FinFET and the TFET implementations as a function of the $V_{DD}$, simulated at various activity factors. As is clearly observed, the TFET outperforms the FinFET at lower $AF$s and $V_{DD}$, with the maximum benefit coming from biasing the circuits at 0.3 V. The benefit in TFET energy efficiency due to reduction in the $AF$ is also seen from Fig. 6c. This is due to a reduction in the contribution of the dynamic energy dissipation component to the total energy dissipation. The TFET performs better than the FinFET for all activity factors below 0.01, and the performance continues to improve as the $AF$ is reduced further. Interesting to note however, is that the energy dissipation in

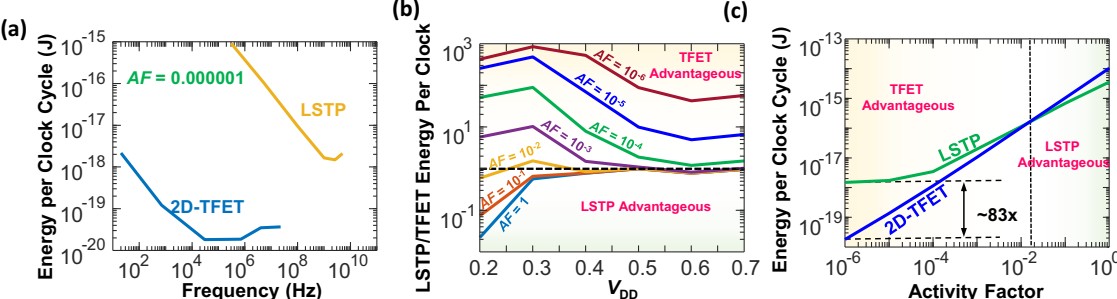

**Fig. 6 | Performance analysis—II.** Comparison of the energy efficiency of the designed NM circuit implemented with 2D-TFET and LSTP circuits. **a** The energy dissipation per clock cycle as a function of frequency plotted for an activity factor of $10^{-6}$. Frequencies correspond to $f_{max}$ and is plotted for corresponding $V_{DD}$. Plot shows tremendous energy-efficiency benefits of implementing NM circuits with TFET w.r.t its LSTP counterpart, with energy efficiencies of several orders of magnitude across the entire frequency range. **b** The ratio of the energy dissipation per clock cycle for the LSTP implementation compared to that of the TFET plotted as a function of $V_{DD}$ with corresponding $f_{max}$ at various activity factors. The horizontal dotted line corresponds to a ratio of 1, thereby implying equal LSTP and TFET energy dissipation. TFET is more advantageous overall, especially for $AF$ below $10^{-2}$, and the energy efficiency increases at lower $V_{DD}$. **c** Energy dissipation per clock cycle plotted as a function of the activity factor plotted by taking the best-case energy dissipation of both TFET and LSTP, corresponding to **b**. The activity factor around which the energy dissipation of the TFET implementation becomes lower than that of the LSTP is shown by a vertical dotted line, showing larger energy efficiencies for 2D-TFETs at lower $AF$s.

the TFET circuit is generally limited by its dynamic energy (since the static energy is relatively smaller), which causes an almost linear change of the energy dissipation with the $AF$. The comparable static and dynamic power dissipation of the FinFET circuit, particularly at smaller $AF$s, causes its energy dissipation to saturate at the lowest $AF$, which is then limited by its static energy. A best-case difference in energy consumption of 83-fold in favor of the TFET is observed at the lowest $AF$.

In summary, the paper introduced and explored a fully functional NM circuit, implemented with 2D lateral-HTJ TFETs, capable of demonstrating a LIF neuron firing operation along with its Hebbian learning circuitry. The circuit was designed to operate at a wide range of supply voltages, ranging from 0.2 V to 0.7 V, and the corresponding maximum frequency of operation ($f_{max}$), static, and dynamic switching energies were extracted to evaluate the total energy consumption of the circuit at a wide range of activity factors. Comparative performance projections carried out against 7 nm LSTP FinFET implementations, under consideration of appropriate interconnect parasitics from the corresponding technology node demonstrated that the 2D-TFET implementation of the NM circuit outperformed the FinFET implementation at activity factors below $10^{-2}$, resulting in a best-case energy dissipation metric of close to two orders of magnitude smaller than that of the latter. The comprehensive analysis and performance projections help introduce an appropriate hardware platform for the implementation of the next generation of high-performance low-energy NM circuits.

## Data availability
The data that support the findings of this study are present in the article and Supplementary Information. Additional data related to this study are available from the corresponding author upon request.

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

## Acknowledgements

The research outlined in this article from the Nanoelectronics Research Lab at UCSB was supported by the Army Research Office (grant W911NF1810366)) and Intel Corporation.

## Author contributions

K.B. initiated the research, organized, and led the collaboration. A.P. carried out the design and analysis of the various circuits with simulation support from Z.C. and J.J., and modeling support from W.C. M.D., V.D. and K.B. supervised the project. A.P. and K.B. wrote the paper with input from all other authors.

## Competing interests

The authors declare no competing interests.
