## [Peer Review File · Nature Communications]

An Ultra Energy-Efficient Hardware Platform for Neuromorphic Computing Enabled by 2D-TMD Tunnel-FETsREVIEWER COMMENTS

Reviewer #1 (Remarks to the Author):

The authors introduce neuromorphic circuits employing two-dimensional layered channel material-based tunnel-field-effect transistors. The tunnel-field-effect transistors characterize low off-state current, small-SS, pristine interface, and suppressed band-tails, among other features. The leaky-integrate-fire (LIF) based digital neuromorphic circuit along with its Hebbian learning circuitry operates at a wide range of supply voltages, frequencies, and activity factors, enabling two orders of magnitude higher energy-efficient computing that is difficult to achieve with conventional material and/or device platforms.

Overall, this manuscript is well written and well organized. And the results will be of interest to the neuromorphic circuit community. This work is appropriate for publication in Nature Communications. However, I have two small issues that I would like to see the authors address before the manuscript is accepted.

What will be the difficulties in manufacturing such a neuromorphic circuit with two-dimensional layered channel material-based tunnel-field-effect transistors?

What will the authors do to further improve the device and circuit performance, addressing issues such as low On-current and high miller capacitance?

Reviewer #2 (Remarks to the Author):

The authors present noteworthy data on low-power 2D-TFET and LSTP digital circuits, and a novel design of a digital LIF neuron and learning synapse.

However they embed this in the context of low-power brain inspired neuromorphic computing, without comparing to the state-of-the-art and making arguable claims about usefulness of this approach for edge-computing (see detailed comments below).

In general, the abstract and the introduction section mix concepts and ideas in ways that do not make sense and are even misleading. For example they justify the benefits of the neuromorphic approach citing the analog-circuit design approach of Carver Mead [1] to claim that neuromorphic enables parallel processing and small footprints, but then propose clocked time-multiplexed (serial) digital circuits which require at least 100x more devices to implement neural and synaptic dynamics. Similarly, they lead to believe that NM will be able to implement "ultra-energy-efficient" systems for AI applications at the edge with "minimal bandwidth requirements", without explaining how this can be achieved, and without clarifying or distinguishing for which applications NM is promising and for which not. Some of the authors themselves in their key-note lectures are very careful to point out that AI and NM applications/technologies are quite different.

As such, the work is of very limited significance to the domain of low-power edge computing (and to that of neuromorphic computing). There is a very large set of literature, ranging from memristive devices, to analog CMOS circuits, to organic- and TFT-electronics that the authors ignore, and which has been proposing alternative and competing designs and approaches for edge-computing tasks (e.g., see Lebanov et al., IEEE TBCAS 2023, or Doremale et al, Nature Electronics 2023, just to cite the two most recent papers that came up from a basic literature search).

The work does not support the generic claims about ultra-low power, made in the title and in the abstract, even though the claim that the 2D-TFET neuron design outperformed a 7nm

FinFET equivalent design is very credible. However, in typical edge-computing applications involving processing of sparse, slow sensory signals, state-of-the-art front-end interfacing devices need to use older technology nodes because of their lower noise and lower leakage, feature. So, at least for the edge-computing devices that integrate also interfaces to sensory signals, the high leakage argument made in the paper is not relevant.

There are also other somewhat misleading claims with respect to robustness and resilience to process variations. Readers are led to believe that having bit-precise digital implementations of NM circuits will lead to efficient large-scale NM computing systems. But real brains use highly heterogeneous analog components and (for a large set of edge-computing applications at least) they clearly outperform digital systems. So implying that digital is better for building large scale (efficient) NM systems can be misleading as this might not be the case. Indeed more and more computational neuroscience studies are coming out highlighting the importance of variability and noise in NM computing systems (e.g., see <https://doi.org/10.1371/journal.pcbi.1010593> , <https://doi.org/10.1101/2023.03.22.533815> , or <https://doi.org/10.1038/s41467-021-26022-3>).

In general the authors should make more precise claims, clarify for which specific application areas or cases their approach can have potential (AI? edge-computing? sparse sensory signals?), and restrict their arguments to those specific cases.

If the authors choose to focus on processing of large (digitized) data-sets for AI applications, perhaps it is even better to drop the whole "neuromorphic" angle and highlight the strengths of their results for what they are, without having to bring in promises of how brain-like computing will solve all the problems of the world.

While there is no disagreement that "the energy dissipation per neuron spike [...] is the most important metric determining the overall efficacy of the circuit implementation" the addition of "or clock cycle" in that sentence is confusing.

Also the previous claim that "The performance of any NM circuit is best given by the energy dissipated per clock cycle" is arguable.

The authors argue that TFETs are highly desirable for NM because NM has low frequency of operation and low activity factors (pg. 3). However intuitively a NM network should not burn significant amounts of power if the neurons are silent or not spiking much (due to their low frequency of operation).

Indeed if the SNN is not spiking (i.e., not computing), then large clock frequencies would appear to lead to "wasted" energy, or low energy efficiency. The energy required to charge up the clock lines and to drive the clock-trees does not seem to be in the overall calculation. The authors should better clarify this aspect.

This is especially important for the "edge-computing" applications in which the data-rate of events that encode real-world sensory signals can be extremely slow (<100Hz).

The methodology is sound for the aspects that are related to circuit analysis, specifications, and performance metrics. However the sections on power-dissipation and energy efficiency are not very clear (see comments above).

Given that energy efficiency is a prominent feature, rather than restricting the analysis to clock cycles and frequencies, the authors should clarify what is the energy per inference, for typical workloads of different use cases. For example in edge-computing applications there might be one inference per second or less. And for data-center AI applications it could be

one every micro-second or less.

More generally, the authors should compare/evaluate their approach with example benchmarks, for AI applications, neuromorphic-friendly applications like the LASSO task, and for signal processing applications at the edge, like ECG, EEG or EMG pattern recognition.

Reviewer #3 (Remarks to the Author):

This paper deals with brain-like energy-efficient computing based on neuromorphic circuits. The authors demonstrate high-performance and low-power consumption by employing novel materials and device structure; 2D layered semiconductors and tunnel FET structures by simulation. They also showed superiority of 2D TFET devices over current CMOS technology. This work presents new direction for next generation high-performance low-energy neuromorphic circuits and therefore should be interesting for many readers.

I suggest the author should consider the following,

Ref [24] couldn't reach ptm.asu.edu. you should avoid referring web site, which may no longer exist. Also Ref [2] and [3].

The section title should be amended from "Results" to "Results and discussion" and from "Discussion" to "Conclusion".

Response to Reviewer Comments for Manuscript NCOMMS-23-44498

Reviewer #1 (Remarks to the Author):

The authors introduce neuromorphic circuits employing two-dimensional layered channel material-based tunnel-field-effect transistors. The tunnel-field-effect transistors characterize low off-state current, small-SS, pristine interface, and suppressed band-tails, among other features. The leaky-integrate-fire (LIF) based digital neuromorphic circuit along with its Hebbian learning circuitry operates at a wide range of supply voltages, frequencies, and activity factors, enabling two orders of magnitude higher energy-efficient computing that is difficult to achieve with conventional material and/or device platforms.

Overall, this manuscript is well written and well organized. And the results will be of interest to the neuromorphic circuit community. This work is appropriate for publication in Nature Communications. However, I have two small issues that I would like to see the authors address before the manuscript is accepted.

Response: The authors thank the reviewer for their meticulous review of the manuscript as well as their constructive feedback. We are delighted to hear that our manuscript is well suited for publication in Nature Communications. In response to the insightful suggestions made, we have made appropriate modifications and clarifications to further enhance its quality. Please find below our response to the two issues raised.

1. What will be the difficulties in manufacturing such a neuromorphic circuit with two-dimensional layered channel material-based tunnel-field-effect transistors?

Response: The authors thank the reviewer for this question. Indeed, the fabrication of circuits utilizing two-dimensional (2D) layered channel material-based tunnel field effect transistors (TFETs) presents specific challenges. Predominantly, these challenges stem from the need for precise area-selective wafer-scale uniform growth of 2D materials and achieving low-resistance contacts, problems that are common to all 2D semiconductor (2DS) channel transistors.

Despite these challenges, the 2DS field has witnessed substantial advancements. Specifically, area-selective large-scale uniform-growth has recently been demonstrated, and breakthroughs in metal-2DS contact engineering by Intel and TSMC have already achieved contact resistance values as low as $200 \Omega \cdot \mu\text{m}$ that satisfy the industry roadmap (IRDS) requirements. These advancements along with other advantages of 2DS are highlighted in a recent high-profile article involving several leaders from the semiconductor industry (*The future transistors*, *Nature* 620, 501–515, 2023), where a strong case for 2DS as the channel-material of choice for next-generation CMOS transistors has been established.

Given these significant advances toward manufacturing of 2DS materials and devices, it is reasonable to argue that our proposed neuromorphic circuit design can be manufactured in large-scale in the near future.

2. What will the authors do to further improve the device and circuit performance, addressing issues such as low On-current and high miller capacitance?

Response: The authors thank the reviewer for this question. Recognizing that low ON-current is an issue in all TFETs, we have explored and proposed the implementation of heterojunction 2D-TFETs, as outlined in our previous work presented at IEDM 2015. This innovative approach can significantly improve the ON-current of the transistors by enhancing the tunneling field at the source-channel barrier. Additionally, we have delved into contact engineering strategies, as evidenced by a recent collaborative work (between UCSB and Intel) at IEDM 2022, aiming to achieve lower resistance contacts through unique contact engineering, which will further contribute to improving the ON – current in scaled 2DS transistors.

On the matter of high Miller capacitance, which can degrade the performance in 2D-TFET based circuits, adoption of a suitable gate-drain underlap at the channel-drain junction will be beneficial. This modification is expected to result in a reduced gate-drain capacitance, thereby mitigating the impact of the Miller capacitance. Furthermore, introduction of this underlap is not expected to lower the ON-current, since the ON-current in TFETs is primarily determined by the tunneling of carriers across the material bandgap at the source-channel junction. Consequently, a gate-drain underlap if kept to a minimal length, should not adversely affect the ON-current.

Finally, these individual device improvements will not only lead to higher-performing devices, but also improve the performance of the circuits.

Reviewer #2 (Remarks to the Author):

The authors present noteworthy data on low-power 2D-TFET and LSTP digital circuits, and a novel design of a digital LIF neuron and learning synapse.

However they embed this in the context of low-power brain inspired neuromorphic computing, without comparing to the state-of-the-art and making arguable claims about usefulness of this approach for edge-computing (see detailed comments below). In general, the abstract and the introduction section mix concepts and ideas in ways that do not make sense and are even misleading.

Response: We thank the reviewer for their thorough review of the paper and the constructive criticisms provided. We have provided a step by step response to the concerns raised below, which should clarify what we intended to convey in the abstract/introduction of the paper.

1. For example they justify the benefits of the neuromorphic approach citing the analog-circuit design approach of Carver Mead [1] to claim that neuromorphic enables parallel processing and small footprints, but then propose clocked time-multiplexed (serial) digital circuits which require at least 100x more devices to implement neural and synaptic dynamics.

Response: We thank the reviewer for this insightful comment. We acknowledge that the original approach used by Carver Mead proposed the use of analog-circuits to achieve neuromorphic computing, emphasizing parallel processing and small footprints. However, nearly all commercial implementations of NM computing in recent years – including Intel’s Loihi, and IBM’s TrueNorth -- utilize digital circuits in scaled process nodes suitable for high-volume manufacturing to exploit the inherent variation/noise-tolerance and easier programmability of digital circuits, as well as affordable monolithic integration of billions of neurosynaptic elements in a compact form factor. Although energy consumption per neurosynaptic element is significantly smaller for analog implementations

and fewer devices are needed per operation, digital implementations offer more flexibility and robust functionality at a system level that must integrate billions of such elements in compact form factors to reduce energy consumptions related to signal transmissions across the elements. We have inserted text in the abstract and introduction sections to further clarify this point.

In citing Carver Mead's paper, our intention was to simply acknowledge that pioneering work, rather than to draw a direct parallel between his approach and ours. Our work specifically aims to demonstrate the advantages of employing two-dimensional (2D) TFET devices over commercially used FinFET architectures for implementing sparse firing NM computing architectures. While we agree with the reviewer that digital circuits, due to their time-multiplexed nature, might necessitate a larger number of devices to implement neural and synaptic dynamics compared to their analog counterparts, the significant energy savings and enhanced performance provided by 2D-TFETs, as demonstrated in our results, offer compelling advantages, making them a viable and attractive option for neuromorphic computing applications. Furthermore, if circuit architecture changes, the energy savings benefits will carry over, and as such the major conclusions of our work will not change.

2. Similarly, they lead to believe that NM will be able to implement "ultra-energy-efficient" systems for AI applications at the edge with "minimal bandwidth requirements", without explaining how this can be achieved, and without clarifying or distinguishing for which applications NM is promising and for which not. Some of the authors themselves in their key-note lectures are very careful to point out that AI and NM applications/technologies are quite different.

Response: We thank the reviewer for this comment. However, please note that the emphasis of this work is on showcasing the energy efficiency and performance benefits of this novel approach, particularly when compared to conventional architectures, that should benefit a wide range of applications. Therefore, the detailed design and implementation of NM circuits for AI applications at the edge is not the purpose of this study. In fact, by demonstrating the fundamental building blocks and their performance characteristics, we aim to provide a solid foundation upon which future work can be performed to realize these applications. We have now removed parts of the text that imply applications of NM computing for general AI systems at the edge with major energy constraints or bandwidth limitations.

3. As such, the work is of very limited significance to the domain of low-power edge computing (and to that of neuromorphic computing). There is a very large set of literature, ranging from memristive devices, to analog CMOS circuits, to organic- and TFT-electronics that the authors ignore, and which has been proposing alternative and competing designs and approaches for edge-computing tasks (e.g., see Lebanov et al., IEEE TBCAS 2023, or Doremaele et al, Nature Electronics 2023, just to cite the two most recent papers that came up from a basic literature search).

Response: We appreciate the reviewer's comments. Our study indeed has a specific focus, which is to demonstrate the potential of 2D-TFETs in digital neuromorphic (NM) circuits, particularly in scenarios with low firing rates. We aim to highlight the energy efficiency and performance benefits derived from this novel material and device structure. Our mentioning of edge computing serves as a contextual backdrop to motivate our research; it is not the central focus of our study. We have taken care to clarify this point in the revised manuscript to prevent any potential misinterpretation.

Regarding the literature the reviewer mentions, we have conducted a comparison of the energy consumption of our implementation with other relevant papers as suggested:

- a. **Lebanov et al., IEEE TBCAS 2023:** The energy consumption per neuron spike for their implementation is estimated to be at 700 pJ. Considering that our 2D-TFET implementation at the most optimum biasing condition dissipates around 2×10^{-20} J each clock cycle, and we need 32 clock cycles for the neuron spike to happen (Figure 4), the energy consumption per neuron spike can be estimated to be around: 6.4×10^{-19} J
- b. **Doremaele et al., Nature Electronics 2023:** This paper does not report any energy/power consumption metrics.
- c. **Gallo et al., Nature Electronics 2023:** While energy consumption per neuron spike is not reported, however, a peak energy efficiency of 0.86 μ J is reported for operations involving 4,080,384 synaptic weights (page 686). Therefore, a simplistic calculation for the total energy consumed per synaptic weight can be estimated to be around 0.21 pJ, which is orders of magnitude higher than our reported energy consumption.
- d. **Frenkel et al., IEEE Trans. Biomed. Circuits Syst. 2019:** The energy per synaptic operation is reported to be 12.7 pJ, which is considerably higher than that reported by us.
- e. **Yin et al., BioCAS 2017:** The energy per classification is reported to be: 51.4 nJ. Although our implementation does not involve any chip-scale demonstration, our energy benefits are expected to be better than this demonstration.
- f. **Zheng et al., ISCAS 2018:** The energy per classification is reported to be: 1.12 μ J. Although our implementation does not involve any classification tasks, our energy benefits are expected to be significantly better than this demonstration.
- g. **Chen et al., IEEE J. Solid-State Circuits 2019:** The energy per synaptic operation is reported to be 3.8 pJ, considerably higher than that reported by us.

Similarly, the energy consumption per synaptic operation for **BrainScale** (Schemmel et al., *Proceedings of 2010 IEEE Int. Symp. Circuits and Sys.*, 1950), **Neurogrid** (Benjamin et al., *Proceedings of the IEEE*, 2014), **TrueNorth** [7], **SpiNNaker** (Furber et al., *Proceedings of the IEEE*, 2014), and **Loihi** [6], are reported to be 174 pJ, 941 pJ, 27 pJ, 27×10^3 pJ, and 105.3 pJ, respectively, as pointed out in this article (Bouvier et al., *ACM Journal on Emerging Technologies in Computing Systems (JETC)*, 2019), which are considerably larger in magnitude w.r.t our implementation. Also, kindly note that all these implementations described above, except in ‘Case a’ (Lebanov et al.), are all digital implementations. Given the focus of our work on demonstrating the implementation of a fundamental neuron and a synapse, a comparison with any chip-scale design is not appropriate, and hence, we have not resorted to citing them in the main paper.

4. The work does not support the generic claims about ultra-low power, made in the title and in the abstract, even though the claim that the 2D-TFET neuron design outperformed a 7nm FinFET equivalent design is very credible. However, in typical edge-computing applications involving processing of sparse, slow sensory signals, state-of-the-art front-end interfacing devices need to use older technology nodes because of their lower noise and lower leakage, feature. So, at least for the edge-computing devices that integrate also interfaces to sensory signals, the high leakage argument made in the paper is not relevant.

Response: We thank the reviewer for this question. Indeed, for typical edge-computing applications involving processing of sparse, slow sensory signals, state-of-the-art front-end interfacing devices may use older technology nodes which have lower leakage than the 7 nm technology nodes employed in this paper. However, while their off-current (I_{OFF}) is lower, their on-current (I_{ON}) is also lower. In fact, as the technology has progressed, so has the ON-OFF ratio of transistors improved with every generation (*The future transistors*, **Nature** 620, 501–515, 2023), and it will be higher for the 7nm

technology nodes w.r.t older technology nodes. Finally, since final performance metric – the energy dissipation, is governed by the ratio of the static power to the frequency of operation, devices with lower OFF- and ON-current will have lower static power and operational frequency. Therefore, a device with poorer ON-OFF ratio will in fact, have a poorer energy consumption metric than that of a device with higher ON-OFF ratio, even if the leakage current of the former is better than that of the latter. Hence, moving to older technology nodes will not be beneficial to the overall energy dissipation of the circuit.

5. There are also other somewhat misleading claims with respect to robustness and resilience to process variations. Readers are led to believe that having bit-precise digital implementations of NM circuits will lead to efficient large-scale NM computing systems. But real brains use highly heterogeneous analog components and (for a large set of edge-computing applications at least) they clearly outperform digital systems. So implying that digital is better for building large scale (efficient) NM systems can be misleading as this might not be the case. Indeed more and more computational neuroscience studies are coming out highlighting the importance of variability and noise in NM computing systems (e.g., see <https://doi.org/10.1371/journal.pcbi.1010593> , <https://doi.org/10.1101/2023.03.22.533815> , or <https://doi.org/10.1038/s41467-021-26022-3>).

In general the authors should make more precise claims, clarify for which specific application areas or cases their approach can have potential (AI? edge-computing? sparse sensory signals?), and restrict their arguments to those specific cases. If the authors choose to focus on processing of large (digitized) data-sets for AI applications, perhaps it is even better to drop the whole "neuromorphic" angle and highlight the strengths of their results for what they are, without having to bring in promises of how brain-like computing will solve all the problems of the world.

Response: We appreciate the reviewer’s concerns and comments on the robustness and resilience of our approach to process variations. However, it is important to clarify that our objective is not to emulate the complexities of the human brain. Instead, we focus on specific aspects of neuromorphic circuits, namely the fundamental functionality of neuronal synapses and firing.

Our claims are tailored to the precise scope of our work. We aim to develop fundamental digital neurons and synapse models and demonstrate their energy efficiency when implemented with 2D-TFETs compared to FinFET-based devices. This focus aligns with our research goals, and we do not intend to create highly heterogeneous systems that aim to replicate the intricacies of the human brain.

Furthermore, we acknowledge that neuromorphic computing has a broad range of potential applications, including AI, edge computing, and processing sparse sensory signals. However, our study narrows down to fundamental digital neuromorphic circuits, and the claims we make are specific to our scope. We emphasize the energy-efficient implementation of these circuits with 2D-TFETs for the applications we address. Finally, in response to this comment, we have now removed all references to ‘edge computing’ and ‘sparse sensory signal processing’ in the current version of the manuscript.

6. While there is no disagreement that "the energy dissipation per neuron spike [...] is the most important metric determining the overall efficacy of the circuit implementation" the addition of "or clock cycle" in that sentence is confusing. Also the previous claim that "The performance of any NM circuit is best given by the energy dissipated per clock cycle" is arguable. The authors argue that TFETs are highly desirable for NM because NM has low frequency of operation and low activity factors (pg. 3).

However intuitively a NM network should not burn significant amounts of power if the neurons are silent or not spiking much (due to their low frequency of operation). Indeed if the SNN is not spiking (i.e., not computing), then large clock frequencies would appear to lead to "wasted" energy, or low energy efficiency. The energy required to charge up the clock lines and to drive the clock-trees does not seem to be in the overall calculation. The authors should better clarify this aspect. This is especially important for the "edge-computing" applications in which the data-rate of events that encode real-world sensory signals can be extremely slow (<100Hz).

Response: We thank the reviewer for their comments on the energy metrics discussed in our paper. In response to the concerns raised, we have made the following adjustments:

1. We have accordingly removed the phrase "or clock cycle in a digital circuit" from the relevant sentence in the latest version of the manuscript.
2. While we acknowledge the importance of energy dissipated per neuron spike as a critical metric for evaluating energy efficiency, we retained the use of energy dissipated per clock cycle in our analysis. This choice is based on the consideration that the spiking behavior of the circuit can vary depending on input conditions, firing thresholds, and other factors. Using energy dissipation per clock cycle provides a more uniform and generalized metric for our analysis. However, for specific applications where energy dissipation per neuron spike is a more relevant measure, it can be calculated by multiplying the energy consumption per clock cycle by the number of clock cycles between neuron spikes, which is 32 in our case. We have now added one line in the revised manuscript in response to this comment.
3. The reviewer raised a valid point regarding circuits with low activity factors, and we acknowledge that low-activity circuits should ideally consume less energy. Nevertheless, it's crucial to emphasize that even in cases of low activity, circuits still dissipate energy in the form of leakage current. This static power component can become a significant portion of the total energy dissipation, as seen in **Fig.6c**. Our motivation for using low-leakage 2D-TFETs in sparse firing neuromorphic circuits is precisely to address this issue. By leveraging 2D-TFETs, we can significantly reduce the overall energy consumption, particularly in scenarios with low activity factors.
4. Our implementation, being a demonstration of a fundamental neuron and synapse, does not have a clock tree, and therefore the energy required to charge up clock trees is not included in the discussion. Furthermore, the absence of a clock tree in our implementation allows us to focus on accurate analysis for low activity factors, where power dissipated during clock cycles can be neglected, as pointed out by the reviewer. We also want to clarify that the energy dissipated by the clock tree, if included, would be similar for both implementations (TFET and FinFET), as their device capacitances are comparable, thereby keeping the conclusions unchanged. Finally, we would also like to point out that although we do not have a clock tree in our implementation (since it is not needed), we have considered accurate interconnect parasitics wherever applicable, to model realistic circuit behavior performance of the simulated neuron, synapse, and systems.
7. The methodology is sound for the aspects that are related to circuit analysis, specifications, and performance metrics. However the sections on power-dissipation and energy efficiency are not very clear (see comments above).

Response: We appreciate the reviewer's comments. We have provided detailed explanations (see response to (6) above) regarding these comments.

8. Given that energy efficiency is a prominent feature, rather than restricting the analysis to clock cycles and frequencies, the authors should clarify what is the energy per inference, for typical workloads of different use cases. For example in edge-computing applications there might be one inference per second or less. And for data-center AI applications it could be one every micro-second or less.

Response: We understand the reviewer's suggestion regarding the computation of energy per inference for different use cases. However, we would like to clarify that our paper primarily focuses on demonstrating the design, functionality, and energy-efficiency/performance of individual fundamental neurons, synapses, and the learning circuitry. It is not intended to present a complete system or real-world implementation for specific applications. Therefore, given the scope of the study, it is not necessary to conduct analyses for energy per inference. However, based on the reviewer's comments we have now added one relevant line in the revised version of our manuscript.

9. More generally, the authors should compare/evaluate their approach with example benchmarks, for AI applications, neuromorphic-friendly applications like the LASSO task, and for signal processing applications at the edge, like ECG, EEG or EMG pattern recognition.

Response: We acknowledge the reviewer's suggestion to compare and evaluate our approach with example benchmarks for various applications. While we appreciate the importance of benchmarking, we would like to clarify that our current work focuses on demonstrating the design, functionality and energy-efficiency of fundamental neurons and synapses. As such, the objectives of this study do not require any comprehensive benchmarking of specific applications or tasks.

Reviewer #3 (Remarks to the Author):

This paper deals with brain-like energy-efficient computing based on neuromorphic circuits. The authors demonstrate high-performance and low-power consumption by employing novel materials and device structure; 2D layered semiconductors and tunnel FET structures by simulation. They also showed superiority of 2D TFET devices over current CMOS technology. This work presents new direction for next generation high-performance low-energy neuromorphic circuits and therefore should be interesting for many readers.

Response: We express our sincere gratitude to the reviewer for their thorough evaluation of our manuscript and the positive feedback provided. We are also very thankful for recommending publication of our paper.

1. I suggest the author should consider the following, Ref [24] couldn't reach ptm.asu.edu. you should avoid referring web site, which may no longer exist. Also Ref [2] and [3].

Response: We thank the reviewer for pointing this out. We have now updated reference [24] with a journal publication, to ensure a more durable and accessible source of information. As for references [2] and [3], although we acknowledge the potential issue with the longevity of webpages, we did not find equivalent data from journal publications, and these references provide relevant and recent market data that is currently best reported on such platforms. Furthermore, the referenced web sources are commonly cited in academia and industry discussions on the subject, and we believe they add value to the context and understanding of the economic potential of neuromorphic computing. We hope the reviewer appreciates our rationale in retaining references [2] and [3] in their current form.

2. The section title should be amended from “Results” to “Results and discussion” and from “Discussion” to “Conclusion”.

Response: We thank the reviewer for this suggestion, and we have accordingly updated the titles of the sections in our latest version of the paper. We believe these changes contribute to a more coherent and reader-friendly structure of our paper and we thank the reviewer once again for their valuable input.

REVIEWERS' COMMENTS

Reviewer #1 (Remarks to the Author):

My concerns have been resolved. It is OK with me to publish this manuscript.

Reviewer #2 (Remarks to the Author):

The authors have addressed almost all the issues raised in a satisfactory way. All the ambiguous or controversial claims have been removed or fixed, and the manuscript sounds much more scientifically sound.

Perhaps to be even more clear about the (revised) focus of the paper the authors should add the keyword "digital" either directly in the title or add it in the third line of the abstract after "improve the energy efficiency of ..."

A minor issue is that the authors write that they removed the phrase "or clock cycle in a digital circuit" from the relevant sentence in the latest version of the manuscript, but it still appears in the latest revision on pg. 11.

There are still few claims that are arguable, but it is perhaps a matter of personal taste. For example,

(1) when the authors claim that SNNs are highly suitable for implementing machine learning (ML) based AI applications they should substantiate the claim with references or explain why/how they can outperform ANNs and dedicated ANN accelerators (including ANN accelerators with dedicated in-memory computing matrix-multiplication HW). To my knowledge this is a very controversial topic with many researchers arguing that SNNs are indeed *worst* than ANNs for typical AI workloads.

(2) when the authors claim that the "most successful commercial implementations" of NM circuits to date are Loihi and TrueNorth, it is not clear how "success" is measured. If it is in terms of sales then this might not be accurate (perhaps Brainchips or GML chips made more sales than TrueNorth). If it is some other metric then the authors should specify it.

(3) the argument that energy can be reduced by increasing the clock frequency is counter-intuitive if one thinks at the system-level and application-level. Intuitively, in a chip that comprises the circuits proposed, but also includes everything else to get the chip to work including clock trees, I/O buffers, registers, etc. increasing clock frequency would increase the system's overall power consumption. An explanation the tradeoffs of high vs low "f" would help in clarifying this.

Having said this, the argument of how comparatively the 2D-TMD Tunnel-FETs circuits proposed would perform better than the pure CMOS alternatives is clear and could be added to show the value of the work proposed.

(4) the discussions on analog vs digital and the arguments put forth by the authors are all valid. However a discussion of how the original approach of Carver Mead of using analog and/or analog+async digital using the 2D-TMD Tunnel-FET devices would compare versus the clocked-digital approach proposed, would be very valuable.

Reviewer #3 (Remarks to the Author):

The authors successfully amended all the parts I had pointed out. This paper is ready for publication in Nature Communications.

Second Response to Reviewer Comments for Manuscript NCOMMS-23-44498

Reviewer #1 (Remarks to the Author):

My concerns have been resolved. It is OK with me to publish this manuscript.

Response: We express our sincere gratitude to the reviewer for their thorough evaluation of our manuscript and for recommending the publication.

Reviewer #2 (Remarks to the Author):

The authors have addressed almost all the issues raised in a satisfactory way. All the ambiguous or controversial claims have been removed or fixed, and the manuscript sounds much more scientifically sound.

Response: We sincerely thank the reviewer for their time in going through the manuscript and are glad to hear that our revisions have addressed most of the concerns. We provide point-by-point responses to the last few recommendations below.

1. Perhaps to be even more clear about the (revised) focus of the paper the authors should add the keyword "digital" either directly in the title or add it in the third line of the abstract after "improve the energy efficiency of ..."

Response: We thank the reviewer for this suggestion, and we have now added the word 'digital' in the third line of the abstract in the final revised version of the manuscript.

2. A minor issue is that the authors write that they removed the phrase "or clock cycle in a digital circuit" from the relevant sentence in the latest version of the manuscript, but it still appears in the latest revision on pg. 11.

Response: We thank the reviewer for pointing this out and we have now removed the phrase from the final revised version of the manuscript.

3. There are still few claims that are arguable, but it is perhaps a matter of personal taste. For example,
 - a. When the authors claim that SNNs are highly suitable for implementing machine learning (ML) based AI applications they should substantiate the claim with references or explain why/how they can outperform ANNs and dedicated ANN accelerators (including ANN accelerators with dedicated in-memory computing matrix-multiplication HW). To my knowledge this is a very controversial topic with many researchers arguing that SNNs are indeed *worst* than ANNs for typical AI workloads.

Response: We thank the reviewer for this suggestion. Although we agree that current ANN networks outperform current SNN networks for ML based AI applications, however, it is expected that the state-of-the-art SNN networks, when realized, will be highly suitable for those

applications too by closely emulating the behavior of the human brain. We have accordingly updated the sentence in the revised version of the manuscript.

- b. When the authors claim that the "most successful commercial implementations" of NM circuits to date are Loihi and TrueNorth, it is not clear how "success" is measured. If it is in terms of sales then this might not be accurate (perhaps Brainchips or GML chips made more sales than TrueNorth). If it is some other metric then the authors should specify it.

Response: We thank the reviewer for this point, and we have accordingly changed the sentence to read: "In fact, some of the most successful commercial implementations of NM circuits to date, such as Intel's Loihi [6] and IBM's TrueNorth [7]". This change conveys that Loihi and TrueNorth represent two examples of the "most successful commercial implementations".

- c. The argument that energy can be reduced by increasing the clock frequency is counter-intuitive if one thinks at the system-level and application-level. Intuitively, in a chip that comprises the circuits proposed, but also includes everything else to get the chip to work including clock trees, I/O buffers, registers, etc. increasing clock frequency would increase the system's overall power consumption. An explanation the tradeoffs of high vs low "f" would help in clarifying this. Having said this, the argument of how comparatively the 2D-TMD Tunnel-FETs circuits proposed would perform better than the pure CMOS alternatives is clear and could be added to show the value of the work proposed.

Response: We thank the reviewer for this point. Referring to **Figure 6a**, which shows that the minimum energy consumption of the circuit is obtained at an intermediate frequency, we would like to clarify that we are plotting the energy consumption of the circuit as a function of the operational voltage, and not the power consumption. Although the reviewer is correct in saying that the operational power, i.e., the dynamic power, of the chip increases as the clock frequency is increased, the dynamic energy consumption is in fact, independent of frequency, and as such would be unaffected by it (unless operational voltage changes). The static energy consumption, however, is inversely related to the operational frequency (Equation 1 in the main manuscript) and would decrease as the frequency is increased. Hence, the overall chip energy consumption would generally decrease as the operational frequency is increased, unless this increase in frequency is obtained from a corresponding increase in operational voltage, in which case both static and dynamic energies would increase linearly and quadratically, respectively. This complex interdependency thus in fact, results in a minimum of chip energy consumption at an intermediate operational frequency. We have now adequately explained this fact on Page 12 of the revised manuscript.

- d. The discussions on analog vs digital and the arguments put forth by the authors are all valid. However a discussion of how the original approach of Carver Mead of using analog and/or analog+async digital using the 2D-TMD Tunnel-FET devices would compare versus the clocked-digital approach proposed, would be very valuable.

Response: We thank the reviewer for this insightful question. The original approach of Carver Mead relied on the use of a voltage controlled current source whose current was related to the input voltage exponentially. Such an exponential behavior of the current on the voltage is in fact obtained from traditional MOS transistors biased in their subthreshold regime where the rate of

change of current with the applied voltage is dictated by its subthreshold slope, inverse of subthreshold swing, given in units of decade/mV. Since for MOS transistors, this subthreshold slope is constant in the entire subthreshold regime, biasing these transistors in this regime yields a consistent change in the output current for a unit change in the gate voltage, thereby facilitating in the design of robust analog NM circuits. However, in TFETs, due to the band-to-band tunneling nature of carriers which causes the carrier tunneling probability to depend non-linearly on the junction electric field in the log-linear scale [see refs. **17,27** in the main manuscript], this subthreshold slope is not constant. Instead, it depends very sensitively on the biasing regime of the transistors. This results in inconsistent change in the output current for a TFET with a unit change in gate voltage, thereby greatly increasing the challenges in the design of robust analog NM circuits and makes the design very variation prone. Therefore, although more energy efficient than conventional MOS transistors, this biasing condition dependent subthreshold slope of the TFETs renders them unsuitable for implementing analog NM circuits in line with Carver Mead's proposal.

Reviewer #3 (Remarks to the Author):

The authors successfully amended all the parts I had pointed out. This paper is ready for publication in Nature Communications.

Response: We would like to express our sincere gratitude to the reviewer for their thorough evaluation of our manuscript and for recommending the publication.